# A Study of the Integrated Model with Norm Activation Model and Theory of Planned Behavior: Applying the Green Hotel's Corporate Social Responsibilities

YoungHyun Kim

Department of Hotel Management, Honam University, Gwangju 62399, Republic of Korea; htlmgr@honam.ac.kr

**Abstract:** This study investigated the structural relationship between the green behavior of green hotel users by integrating and applying green corporate social responsibility (GCSR), the norm activation model (NAM), and theory of planned behavior (TPB). The survey targets were customers who have visited green hotels at least 1–2 times in the past two years and completed an online (mobile) self-written Google survey. In addition, to increase the validity of the research hypothesis of this study, the definition of green hotel CSR was simply explained for understanding before the survey began. Four hundred and fifty surveys were distributed for a month from 15 October 2022, and 386 samples were used for final analysis, excluding non-response, and SPSS and Amos programs were used for the analysis. The analysis results of the research hypothesis are as follows. In the process of GCSR on the green behavior of hotel users, the relationships between awareness of consequence (AC), ascription of responsibility (AR), and personal norm (PN) of NAM were all found to have a significant effect. Therefore, H1a–d are supported. Additionally, in the context of another theory, TPB, in the process of GCSR on the green behavior of hotel users, the study verified the relationship between attitude (AT), subjective norm (SN), and perceived behavioral control (PBC) of TPB. However, GCSR has no significant effect on PBC, so H2c was rejected, and PBC has no significant effect on green behavioral intention (GBI), so H4c was rejected. Finally, subjective norm (SN) as a variable of TPB was found to have a significant positive effect on PN as a variable of NAM. Thus, H3 was supported. This study intends to present useful basic evidence by providing academic and practical implications for establishing the management strategy of hotel companies.

**Keywords:** green corporate social responsibility; awareness of consequence; ascription of responsibility; personal norm; attitude; subjective norm; perceived behavioral control; green behavioral intention

## 1. Introduction

As consumer interest in social and environmental issues such as global warming and environmental pollution increases, the social perceptions of providing goods and services that do not harm the human body have changed while minimizing the impact on nature [1,2]. In addition, as consumer interest in ethical transparency, social performance, and the environment increases, the importance of corporate social responsibility (CSR) is increasingly emphasized [3]. For example, the types of CSR being implemented by companies are as follows: utilizing renewable energy and reducing the use of fossil fuels (oil, coal, natural gas) to reduce carbon dioxide emissions [4]; reducing industrial waste and minimizing plastic usage (food, textiles, construction, and plastic packaging) [5]; simplifying reports (printers, ink, etc.) for paperless offices; and manufacturing high-efficiency products to save energy [6].

As such, global companies' management operational strategies are becoming more environmentally friendly, and there are many efforts being made to strengthen the corporate image of eco-friendly activities and CSR in terms of companies' competitive advantage. In the past, social responsibility simply involved donation activity to improve the image of a

company and pursue profit. However, it is now considered important as a strategic activity that can lead to continuous growth by linking and implementing various responsibilities demanded by society [7].

In this context, there are various movements toward environmentalism in the tourism industry. Hotels are implementing eco-friendly programs in their rooms, building eco-friendly buildings, and introducing and adopting programs to save water, energy, and reduce waste [8]. In addition, eco-friendly hotels are encouraging customers to participate in their activities, and through voluntary participation of hotel guests, bed sheets, bathroom towels, and shower gowns are reused during the stay, thereby reducing water pollution, and preserving the global environment [9,10]. Environmentalism and CSR can help customers to recognize higher values in the process of being given the opportunity to participate [11,12], along with second visits with an open mind about the additional costs that customers have to pay [13] and positive thoughts toward eco-friendly companies, resulting in increased sacrifice for the environment [14].

Accordingly, the CSR of eco-friendly hotel companies is evaluated positively by customers and has a significant impact on customers' attitudes and behaviors [11]. In previous studies [10,15–18], customers using eco-friendly companies with CSR showed an increase in eco-friendly behavior. However, explaining various customer behaviors is limited when their behavior is viewed simply as eco-friendly behavior. This is because customers should bear the premium costs; it is also essential to examine in detail the willingness to revisit, eco-friendly consumption intentions, and habits that arise as a result of their experience in eco-friendly hotels. To predict this specific behavior, it is fundamental to examine people's perceptions and norms of responsibility toward eco-friendliness, which have been proven in studies using the norm activation model (NAM) [11,13,19,20].

The NAM is a model that addresses the limitations of the theory of planned behavior (TPB)—used widely in consumer behavior research, including tourism—in which humans are assumed to be rational beings and not driven by individual internal norms [21]. It is considered suitable for efficient prediction in relation to the psychological state of customers according to the hypothesis presented in this study. Accordingly, in this study, we tried to predict the behavioral intention of eco-friendly hotel customers by applying the NAM as the cause variable of green CSR (GCSR). Moreover, according to the TPB, attitude (AT), subjective norms (SNs), and perceived behavioral control (PBC) have also been proven to have a significant influence on behavioral intentions through a number of previous studies [22]. In addition, the TPB is a social cognitive theory that predicts behavior and has been widely used in research on eco-friendly behavior [23]. However, while the TPB is useful for explaining and predicting human behavior, additional variables are needed to increase its explanatory context because the theory's explanatory content for behavioral intent is only 30% to 40%. Given that the TPB predicts behavioral intentions based on rational judgment, the necessity to consider not only subjective norms (SNs), which are social norms, but also moral norms such as personal norms (PNs), is emphasized [19,21,23,24].

Regarding green hotels, a number of studies [12,24–28] applied the TPB to verify the influence relationship of green behavior, sacrifice reduction intention, and premium payment intention, while [11,19,29–31] applied the NAM to demonstrate a significant influence relationship with people's green behavior intention. As such, the TPB and NAM, which are known to efficiently predict green behavior, have been studied with various approaches [19,23,32], but the hypothetical approach through the integrated framework of the TPB and NAM is very limited. Despite the research results that a green CSR image has high positive results for corporate financial and non-financial management performance and sustainable management [25,33], research in the hospitality industry is very insufficient.

Although they are founded on different theoretical models, the two discussed theories—the TPB and NAM—share the objective of ultimately determining green behavioral intentions. In this study, we looked at the perception of environmental problems of the NAM in the context of green CSR images, examining attitude and positive effect. People who are aware of environmental problems are more likely to have a positive attitude toward

eco-friendly activities [34,35]. The subjective norm for self-environmental behavior, on the other hand, is socially desirable and serves as a guide for the personal norm that acknowledges it [19]. As a result, the subjective norm for the TPB positively affects the norm for the NAM.

The first purpose of this study is to verify the influence relationship between GCSR and awareness of consequence (AC), ascription of responsibility (AR), PN, and green behavioral intention (GBI) with empirical analysis. Second, the relationship between GCSR and AT, SN, PBC, and GBI is verified. Third, by confirming the relationship of SN and PN, comparative verification of the influence on the NAM and TPB is undertaken. Therefore, through the analysis, we intend to present basic data for the establishment of efficient management strategies for eco-friendly hotel companies.

## 2. Theoretical Background

### 2.1. Green Hotels, Green CSR of Green Hotels, and Green Hotel Cooperation

During the 1930s, CSR first appeared in the United States. In the seminal text "Social Responsibilities of the Businessman", Bowen emphasized the obligation of desirable actions, judgments, and principles for social goals and values [36]. The hotel industry accounts for a large portion of environmental pollution, whereby industries are classified by their $CO_2$ emissions. According to the Environmental Protection Agency, the hotel industry is included in the top five sectors for $CO_2$ emissions in the United States, requiring urgent countermeasures [37]. An eco-friendly hotel is defined as a hotel actively operated by hotel operators in implementing programs to protect people and the planet as well as contributing to economic benefits by saving water and energy and reducing the amount of waste [38]. It can also be defined as an active organization that implements environmental protection policies and practices by adding social responsibility to the pursuit of short-term interests [39,40]. In other words, hotels that run eco-friendly programs by implementing the goal of minimizing their negative impact on the environment in their business vision can be considered eco-friendly hotels [41].

It is suggested that eco-friendly management is necessary so that customers recognize and evaluate the image of a company in various ways related to eco-friendliness; this is achieved by introducing eco-friendly programs based on legislation and reducing operating costs as well as maintaining competitiveness and the introduction of quality assurance systems, energy management, water quality protection, and waste management measures [42]. For example, in the case of a hotel company with an eco-friendly mark, the hotel can show an image of GCSR, portraying itself as a hotel that takes responsibility for environmental issues and applies many eco-friendly elements to management activities without changing the basic facilities and services available to promote a glamorous and luxurious brand [43,44]. In addition, they are more interested in managing the long-term brand image as a company that fulfills social responsibility by actively reflecting environmental preservation policies, away from the goal of pursuing short-term profits of existing hotel companies. As a result, customers think positively of the image of a hotel company that performs GCSR [45,46].

Therefore, the GCSR of an eco-friendly hotel is defined as when a hotel implements eco-friendly policies, or green management activities, without lowering the value or quality of existing services with a sense of social responsibility for environmental protection [25,33]. The purpose of eco-friendly management is to preserve natural and cultural resources, and to pursue profits for existing companies by revitalizing the local community. This occurs by making not only customers but also employees responsible for the environment [47]. The environmental protection activities and green management programs pursued by hotel companies can receive positive responses from customers who care about the environment and are important consumers in purchasing and using hotel products [19,26,48].

Hotel companies' CSR therefore has an important influence on the formation of the corporate image and consumers can show positive effects on the company by having a sense of responsibility, interest, and motivation for the environment. In addition, consumers play a positive role in corporate image, product, and service image together with positively assessing companies engaged in CSR [49].

*2.2. Relationship between Corporate Social Responsibility, the Norm Activation Model, and Green Behavioral Intention in Green Hotels*

Recently, consumers have shown greater interest in environmental issues, and customers who purchase eco-friendly products or prefer related actions demonstrate a sense of responsibility for the environment, leaning toward eco-friendly consumption behavior, or eco-friendly behavior, with a will to contribute to solving environmental problems [50].

Accordingly, Schwartz (who first introduced the NAM in a study on eco-friendly behavior), particularly emphasized PNs among the variables constituting the model [29]. Through his work, PNs have been found to have a direct influence that causes eco-friendly behavior with a sense of moral obligation [51]. Thus, PNs are mainly activated by the following situational variables: first, AC concerns the perception of negative consequences that occur when a person does not act in an eco-friendly way toward another person or a particular situation. Second, AR refers to the responsibility for the negative consequences of not acting in an eco-friendly way. Since then, the NAM has been recognized as a useful model for explaining prosocial behavior, and most related studies use the three variables that Schwartz first proposed: AC, AR, and PN [52]. The NAM has since been considered a useful model for explaining eco-friendly behavior, and demonstrated in prior studies of the hospitality industry, such as customer revisit intention toward green hotels [13], environmentally responsible behavior of smart tourists [53], environmentally responsible decision-making processes of cruise passengers [54], and environmental decision-making processes of hotel users [55]. In conclusion, analysis shows that PNs are created by individuals' awareness of the environment and their responsibility to solve problems, and that if people can recognize the seriousness of environmental problems, they are likely to take eco-friendly actions [56]. For example, the management activities of eco-friendly companies (hotels/resorts/cruises) create awareness and responsibility for environmental issues, which increases customers' eco-friendly behaviors, such as saving electric energy and water, reusing towels and linen, reducing waste, and recycling [57–61].

The NAM proposed by Schwartz [29] explains behavioral intentions based on moral norms, and [13,44,62] proved the causal relationship of the consumer decision-making process through the NAM in the context of green food, tourism, and hotel industries, respectively. In this context, Han et al.'s study [11] of airlines' and restaurants' GCSR confirmed a significant positive relationship between individual moral norms and eco-friendly behavior [11]. In addition, Han et al.'s research [63] found that improving airline image using a GCSR strategy could enhance customer loyalty. It is therefore evident that the relationship between the hospitality industry's corporate image and NAM variables is closely correlated with customers' eco-friendly behavior [19]. To summarize, eco-friendly hotel companies need to establish an eco-friendly image in various ways to instill awareness and responsibility for environmental problems in customers, which can lead to more effective customer eco-friendly behavior.

This framework originally postulated that an individual's general ecological worldview—measured by the new ecological paradigm [64]—determines AC. In turn, this awareness increases AR. Support for a positive relationship between worldview and awareness and between awareness and responsibility has been provided in various contexts. Therefore, the aim of this study, based on the previously mentioned studies, is to confirm the relationship between AR, AC, PN, variables of the NAM, and eco-friendly behavioral intention as per [29–31] and the GCSR of eco-friendly hotels as per [11,19]. Accordingly, the following hypotheses were established:

**Hypothesis 1a (H1a).** *The CSR of green hotels will have a significant effect on AC.*

**Hypothesis 1b (H1b).** *The AC will have a significant effect on AR.*

**Hypothesis 1c (H1c).** *The AR will have a significant effect on PN.*

**Hypothesis 1d (H1d).** *PN will have a significant effect on GBI.*

*2.3. The CSR of Green Hotels and the TPB*

Discussions on sustainable development related to the environment have changed the needs and behaviors of consumers using products and services. Consequently, there is an increased interest in consumer behavior in both industrial practice and academic research contexts [65–68]. Accordingly, research is needed to support consumers who participate in companies that implement eco-friendly policies with responsibility and take practical actions, as well as to understand the behavioral process of consumers visiting eco-friendly companies.

Research on the eco-friendly behavior of consumers has been continuously conducted and expanded with the increase in environmental problems [69]. Consumers have a positive view of companies with GCSR and it has a positive effect on overall corporate image. Caudron [70] stated that the corporate image recognized by consumers through corporate social activities can be an important criterion for purchasing decisions, and the consumer's perception of whether a company responds well to consumer needs is an advantage for companies, while commercial profit is a weakness. In other words, the image of a company can change customers' perceptions, play an effective role in changing attitudes and behaviors [71,72], and increase eco-friendly behavior through customers' positive perceptions by aiming for high GCSR [73,74].

In this context, this study attempted to investigate the structural relationship by applying the TPB, which is recognized to be the most efficient way to predict consumers' eco-friendly behavior. First, the TPB is a model developed by expanding the theory of reasoned action [75]. The TPB explains individual willful behavior by human rationality, although in reality, there are many things that are beyond individual control and there is a limit to explaining these things only with the TRA. Moreover, the actions that actually take place require money, time, opportunity, and resources, but these are not all voluntarily controlled by individuals [76]. To compensate for this problem, this theory considers PBC as a prerequisite for behavioral intentions as well as behavioral attitudes and SN, recognizing that it directly affects behavior as well as behavioral intension [75,77]. In addition, AT and PBC in the TPB are factors that explain individual characteristics, and SN can be regarded as social factors. Thus, AT, SN, and PBC are explanations and predictable factors for behavioral intentions in an individual's decision-making process [75]. Because there are obstacles or uncertainties in targeting a particular action that an individual is trying to achieve, these factors must be considered to an extent, which is the role of PBC [78]. In other words, PBC is a concept that encompasses the concepts of self-efficacy and controllability, which mean the degree to which an individual's actions are being carried out within their control [75]. Therefore, even if consumers have a favorable AT and SN in terms of their behavior, if they have less control over the behavior they perform, the intention to act is reduced [79]. Next, norms are understood to be an important and easy-to-use concept in relation to behavior and have been reported as an important factor concerning consumers' change toward eco-friendly behavior [80]. In addition, if an individual agrees that norms guide people to perform eco-friendly behaviors and that they are important for society, they will perform eco-friendly behaviors themselves as a result of social pressure [81–83].

Corporate social responsibility is increasingly important for driving consumers' daily consumption decisions. A growing number of hotels and restaurants have engaged in numerous CSR initiatives to conserve natural resources, save costs, and enhance employee and consumer loyalty [84]. Previous research shows that consumers' own views about

social and environmental issues and their perceptions of firms implementing CSR programs have a significant impact on their behavioral intentions [85,86].

In this context, its importance and validity have been verified in a number of prior studies [12,24–28] in the tourism industry applying the TPB to predict customers' eco-friendly behavioral intentions. Specifically, a study of Vietnamese shopping mall customers verified the significant relationship between environmental CSR and eco-friendly product purchase intentions [87]. Consequently, it was found that green CSR had a high correlation with AT, SN, and PBC. In addition, green CSR had a stronger influence on women's intention to purchase eco-friendly products than on men. In addition, a study of airline passengers and restaurant customers showed the significant influence of attitudes toward environmental CSR and eco-friendly products on both participant groups [11]. In addition, it was confirmed for the purchasing power that airline passengers had a higher influence towards eco-friendly products than restaurant customers. It was also confirmed that SN and PN had a high correlation through a number of prior studies [19,23,88–91].

Accordingly, based on prior studies that empirically analyzed a combination of the TPB and the NAM [19,23,32], this study attempted to further confirm the influence of GCSR on each sub-factor of the TPB and the influence of SN of the TPB on PN of the NAM. Therefore, the following hypotheses were established:

**Hypothesis 2 (H2).** *GCSR has a significant effect on the TPB.*

**Hypothesis 2a (H2a).** *GCSR has a significant effect on AT.*

**Hypothesis 2b (H2b).** *GCSR has a significant effect on SN.*

**Hypothesis 2c (H2c).** *GCSR has a significant effect on PBC.*

**Hypothesis 3 (H3).** *SN has a significant effect on PN.*

**Hypothesis 4 (H4).** *The TPB has a significant effect on GBI.*

**Hypothesis 4a (H4a).** *AT has a significant effect on GBI.*

**Hypothesis 4b (H4b).** *SN has a significant effect on GBI.*

**Hypothesis 4c (H4c).** *PBC has a significant effect on GBI.*

## 3. Method

### 3.1. Measures and Questionnaire Development

This study focused on (1) expected customers' green behavioral intention of visiting a green CSR hotel, (2) applying the norm activation model (NAM) and theory of planned behavior (TPB), (3) the utility of the NAM and TPB as verified through previous studies of environmental science [13,15–17,54,55,92]. With these goals, we intended to accumulate studies of hotel industry hypotheses in the previous studies mentioned above with a research model based on the presented hypotheses and schematized it as Figure 1.

Measurement variables of previous studies examined the change in customer intentions against green corporate social responsibility (GCSR). However, there were insufficient studies related to the hotel industry, as most of the variables were from studies of the tourist industry [9,18,93–95]. Thus, we developed a measurement variable based on modified previous study material and in-depth interviews carried out with hotel managers. In detail, we first selected seven hotel managers with over ten years of experience and five professors of hotel management who were willing to join the in-depth interview. Second, we made detailed explanations of the purpose of the study and interviews. Third, to check the suitability of the questionnaire, the questionnaire used in a previous study was shown to the experts through a one-on-one interview, and it was verified whether the questionnaire

was suitable for proenvironmental hotel research. Fourth, by combining five interviews with experts, it was modified and supplemented to fit the environmental hotel study. To reduce the CMB, the questionnaire was designed to protect respondent anonymity and reduce evaluation apprehension [96,97].

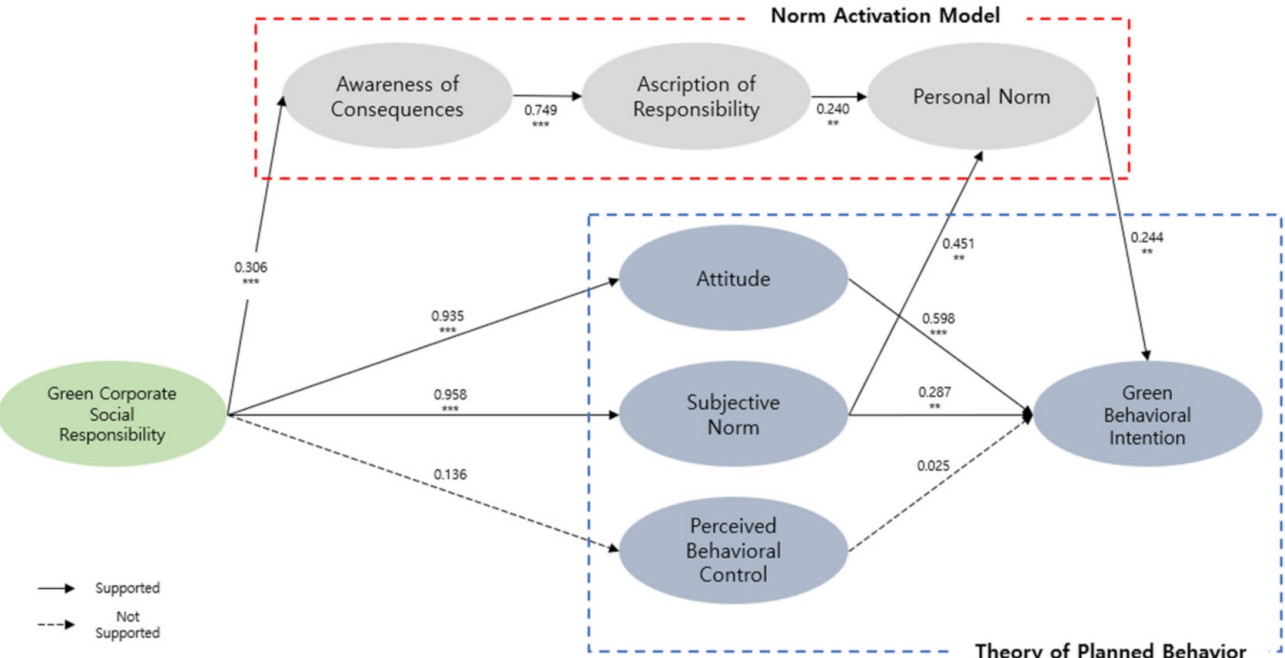

**Figure 1.** Note 1. Green corporate social responsibility (GCSR), awareness of consequence (AC), ascription of responsibility (AR), personal norm (PN), attitude (AT), subjective norm (SN), perceived behavioral control (PBC), green behavioral intention (GBI); Note 2. Goodness-of-fit statistics for the measurement model: $\chi^2$ = 577.441, df = 318, $p$ < 0.001, $\chi^2$/df = 1.816, RMSEA = 0.040, CFI = 0.937, IFI = 0.938, TLI = 0.925, GFI = 0.993; Note 3. Goodness-of-fit statistics for the structural model: $\chi^2$ = 600.793 df = 329, $p$ < 0.001, $\chi^2$/df = 1.826, RMSEA = 0.041, CFI = 0.934, IFI = 0.935, TLI = 0.924, GFI = 0.992, ** $p$ < 0.01, *** $p$ < 0.001.

The operational definition of the variables used in the analysis is as follows. Green corporate social responsibility (GCSR) included 5 items as a single factor by [98]. For AC, AR, and PN of the norm activation model, a total of 10 items were used: 4 items of AC, 3 items of AR, and 3 items of PN based on those used in [44,54,99]. AT, SN, and PBC of the theory of planned behavior were developed by [75,100], and a total of 9 items were used for each of the 3 items used in [26], which surveyed hospitality industry customers. Next, for green behavioral intentions, 4 items were used from [101]. Demographic questions consisted of five questions on: gender, age, marital status, education level, and the number of visits to a green hotel.

Lastly, we deployed the survey to twenty customers who had visited a green hotel, in Korea, from 15 September 2022 for a month as a pretest under final review, providing transparency to the survey. This study used 29 items to evaluate 8 variables, using a five-point Likert scale: 1. Not important at all, 2. Not important, 3. Neutral, 4. Important, 5. Very important.

### 3.2. Data Collection Process and Demographic Profiles

This study comprised a survey to verify the relationship between GCSR and AC, AR, PN, AT, SN, PBC, and GBI, focusing on users of eco-friendly hotels in Korea. The hotel industry in Korea has been bringing positive changes to the service market due to the green consumption craze and ESG management strategy. Almost all five-star green hotels in

Seoul, the survey area of this study, have been certified in accordance with the criteria of Leadership in Energy and Environmental Design (LEED).

The data collection period was one month from 15 September to 15 October 2022, with the survey conducted online (mobile) using a Google survey for users who have visited an eco-friendly hotel in Korea at least 1–2 times in the past two years. After distributing 450 questionnaires, 386 were returned and used for the final empirical analysis (after excluding insincere or non-responses). Beforehand, initiatives of eco-friendly hotels and hotel companies carrying out GCSR were explained. Subsequently, only respondents with a Likert-scale response of 4 points or higher for "degree of awareness of eco-friendly hotel CSR" were used for the analysis. Convenience store beverage coupons were provided to the respondents for their active participation.

Regarding the demographic characteristics of the 386 participants, 52.07% (n = 201) were female customers and 47.93% (n = 185) were male customers. Moreover, 59.84% (n = 231) of the respondents were single while 40.16% were unmarried (n = 155). In addition, 30.83% (n = 119) of respondents were 20–29 years old, 26.68% (n = 103) of respondents were 30–39 years old, 23.83% (n = 92) of respondents were 40–49 years old, 15.54% (n = 60) of respondents were 50–59 years old, and 3.11% (n = 12) of respondents were 60 years old or above. Their levels of education were high school graduate or below (7.77%, n = 30), 2-year college (41.19%, n = 159), undergraduate (34.20%, n = 132), and graduate or higher (16.84%, n = 65). The number of visits to a green hotel were 1~2 times (51.30%, n = 198), 3~4 times (37.82%, n = 146), 5 times or more (10.88%, n = 42).

## 4. Results

### 4.1. Confirmatory Factor Analysis

Before conducting a confirmatory factor analysis (CFA), data screening was performed to check whether there were any violations of the assumptions. First, the common method bias (CMB) was checked by Harman's single-factor and common latent factor approaches. All measurement items were loaded into one common factor and the total variance for a single factor was less than 50% (48.23%). There were no significant differences between CFA with and without the common latent factor when comparing standardized regression weight. Therefore, CMB may not have occurred in our study [102].

The study's measurement model was generated by conducting a confirmatory factory analysis. The confirmatory factor analysis (CFA) was performed to verify the reliability and validity. As a result of the measurement model, goodness-of-fit statistics for the measurement model, $\chi^2 = 577.441$, df = 318, $p < 0.001$, $\chi^2/\text{df} = 1.816$, RMSEA = 0.040, CFI = 0.937, IFI = 0.938, TLI = 0.925, GFI = 0.993, were judged to be excellent overall [103]. Factor loadings, significance probability of $t$-value, average variance extracted (AVE), and construct reliability (CR) were checked to check the convergent validity of the latent variables of the measurement model. The confidence coefficients (Cronbach's $\alpha$) of factor loading were between 0.592 and 0.928, which were more significant than the 0.5 suggested by [104]. Moreover, AVE values and CR values were constructed ranging from 0.512 to 0.741 and from 0.607 to 0.894, respectively. These values were all greater than the levels of 0.5 and 0.7 suggested by [105].

In addition, correlation analysis was performed as shown in Table 1 to verify discriminant validity. As a result of Pearson's correlation analysis, all variables of GCSR, AC, AR, PN, AT, SN, PBC, and GBI were $p < 0.05$, indicating a significant correlation association [106]. Thus, discriminant validity was confirmed.

**Table 1.** The measurement model and correlation.

| Construct and Scale Item | | Standardized Loading | Mean (SD) | AVE (CR) | GCSR | AC | AR | PN | AT | SN | PBC | GBI | $\sqrt{\text{AVE}}$ |
|---|---|---|---|---|---|---|---|---|---|---|---|---|---|
| GCSR | GCSR1 GCSR2 GCSR3 GCSR4 GCSR5 | 0.787 0.824 0.828 0.698 0.694 | 4.06 (0.520) | 0.587 (0.881) | 1 | | | | | | | | 0.766 |
| AC | AC1 AC2 AC3 AC4 | 0.758 0.718 0.622 0.573 | 3.95 (0.587) | 0.533 (0.804) | 0.634 *** | 1 | | | | | | | 0.730 |
| AR | AR1 AR2 AR3 | 0.829 0.741 0.783 | 3.75 (0.706) | 0.611 (0.837) | 0.447 *** | 0.639 *** | 1 | | | | | | 0.782 |
| PN | PN1 PN2 PN3 | 0.885 0.763 0.582 | 4.07 (0.582) | 0.579 (0.777) | 0.523 *** | 0.571 *** | 0.556 *** | 1 | | | | | 0.761 |
| AT | AT1 AT2 AT3 | 0.638 0.641 0.630 | 4.02 (0.544) | 0.597 (0.663) | 0.346 *** | 0.449 *** | 0.614 *** | 0.630 *** | 1 | | | | 0.773 |
| SN | SN1 SN2 SN3 | 0.650 0.622 0.538 | 3.98 (0.523) | 0.558 (0.607) | 0.493 *** | 0.413 *** | 0.392 *** | 0.494 *** | 0.280 *** | 1 | | | 0.747 |
| PBC | PBC1 PBC2 PBC3 | 0.776 0.892 0.913 | 3.84 (0.547) | 0.741 (0.894) | 0.540 *** | 0.496 *** | 0.483 *** | 0.558 *** | 0.373 *** | 0.599 *** | 1 | | 0.861 |
| GBI | GBI1 GBI2 GBI3 GBI4 | 0.619 0.739 0.746 0.615 | 3.56 (0.633) | 0.512 (0.724) | 0.241 *** | 0.327 *** | 0.370 *** | 0.361 *** | 0.465 *** | 0.225 *** | 0.259 *** | 1 | 0.716 |

Note 1. SD = standard deviation, AVE = average variance extracted, CR = composite reliability, GCSR = green corporate social responsibility, AC = awareness of consequence, AR = ascription of responsibility, PN = personal norm, AT = attitude, SN = subjective norm, PBC = perceived behavioral control, GBI = green behavioral intention; Note 2. Goodness-of-fit statistics for the measurement model: $\chi^2$ = 577.441, df = 318, $p < 0.001$, $\chi^2/\text{df}$ = 1.816, RMSEA = 0.040, CFI = 0.937, IFI = 0.938, TLI = 0.925, GFI = 0.993, *** $p < 0.001$; Note 3. All factor loadings are significant at $p < 0.001$.

*4.2. Structural Model and Hypothesis Testing*

In this study, the GBI of green hotel customers was investigated based on the NAM and TPB. The structural equation model (SEM) analysis was generated by using the maximum likelihood estimation method as an estimation method for both model and procedure evaluation [103]. Goodness-of-fit statistics for the structural model ($\chi^2$ = 600.793 df = 329, $p < 0.001$, $\chi^2/\text{df}$ = 1.826, RMSEA = 0.041, CFI = 0.934, IFI = 0.935, TLI = 0.924, GFI = 0.992) were satisfactorily higher than the standard value.

Moreover, the SEM showed high prediction power for $R^2(\text{AC})$ = 0.706, $R^2(\text{AR})$ = 0.735, $R^2(\text{PN})$ = 0.466, $R^2(\text{AT})$ = 0.630, $R^2(\text{SN})$ = 0.610, $R^2(\text{PBC})$ = 0.310, $R^2(\text{GBI})$ = 0.585 and *t*-values and standardized path coefficient are shown in Table 2. The path estimates show that GCSR had a significantly positive effect on AC ($\beta$ = 0.306, t = 8.336 ***). AC had a significantly positive effect on AR ($\beta$ = 0.749, t = 11.186 ***), AR had a significantly positive effect on PN ($\beta$ = 0.240, t = 2.864 **). PN had a significantly positive effect on GBI ($\beta$ = 0.244, t = 2.973 **). Thus, H1a, H1b, H1c, and H1d were supported. GCSR had a significantly positive effect on AT ($\beta$ = 0.935, t = 6.921 ***). GCSR had a significantly positive effect on SN ($\beta$ = 0.958, t = 6.708 ***). GCSR had a significantly positive effect on PBC ($\beta$ = 0.136, t = 1.432). Thus, H2a and H2b were supported, but H2c was not supported. SN had a significantly positive effect on PN ($\beta$ = 0.451, t = 2.805 **). Thus, H3 was supported. AT had a significantly positive effect on GBI ($\beta$ = 0.598, t = 4.583 ***). PN had a significantly positive effect on GBI ($\beta$ = 0.287, t = 2.327 **). PBC had a significantly positive effect on GBI ($\beta$ = 0.025, t = 0.482). Thus, H4a and H4b were supported, but H4c was not supported.

**Table 2.** Hypothesis testing.

| Hypothesized Paths | Coefficients | *t*-Values |
|---|---|---|
| H1a: GCSR → AC | 0.306 | 8.336 *** |
| H1b: AC → AR | 0.749 | 11.186 *** |
| H1c: AR → PN | 0.240 | 2.864 ** |
| H1d: PN → GBI | 0.244 | 2.973 ** |
| H2a: GCSR → AT | 0.935 | 6.921 *** |
| H2b: GCSR → PN | 0.958 | 6.708 *** |
| H2c: GCSR → PBC | 0.136 | 1.432 |
| H3: SN → PN | 0.451 | 2.805 ** |
| H4a AT → GBI | 0.598 | 4.583 *** |
| H4b: PN → GBI | 0.287 | 2.327 ** |
| H4c: PBC → GBI | 0.025 | 0.482 |
| **Explained variable:** | $R^2$(AC) = 0.706, $R^2$(AR) = 0.735, $R^2$(PN) = 0.466, $R^2$(AT) = 0.630, $R^2$(SN) = 0.610, $R^2$(PBC) = 0.310, $R^2$(GBI) = 0.585 | |

Note 1. Green corporate social responsibility (GCSR), awareness of consequence (AC), ascription of responsibility (AR), personal norm (PN), attitude (AT), subjective norm (SN), perceived behavioral control (PBC), green behavioral intention (GBI); Note 2. Goodness-of-fit statistics for the structural model: $\chi^2$ = 600.793 df = 329, $p < 0.001$, $\chi^2$/df = 1.826, RMSEA = 0.041, CFI = 0.934, IFI = 0.935, TLI = 0.924, GFI = 0.992, ** $p < 0.01$, *** $p < 0.001$.

## 5. Discussion

### 5.1. Discussion and Implications

This study has theoretical implications as the model that integrates the NAM and the TPB based on theoretical consideration was identified in the context of green hotels. In addition, the factors influencing customers' green behavior toward green hotels were studied through a model in which the two theories were integrated in the context of the hospitality industry. The purpose of the study was to examine the influence of the predictors AC, AR, and PN presented by the NAM, which were applied in the study as per [11,19,29–31] and the predictors AT, SN, and PBC presented by the TPB as per [12,26–28,48] on the intention to act on eco-friendly hotels. First, considering the need for an integrated approach between the TPB and the NAM, prior studies applying the TPB were evaluated to have insufficient influence to predict customer behavioral intentions, whereas in this study, the analysis of applying the TPB and the NAM together showed that the overall influence has increased. In addition, most of the hypotheses in this study supported the results of prior studies, although PBC did not directly affect GBI.

This study applies the NAM to the customers' green attitude and behavior. Han et al.'s study [11] to predict sustainable consumption and the research results of [30] using the NAM focusing on awareness, responsibility, and norms demonstrated the association between green CSR and the NAM emphasized in this study. In addition, it was found that the research results of [88–91] and the results of this study were supported by the relationship between the perception and behavioral intention of hotel customers by applying the extended TPB. Unlike the research results of [19,23,88], in this study, PBC did not directly affect GBI. However, SN proved its importance from a theoretical integration perspective as a predictor of PN and GBI formation. The specific theoretical and practical implications of the results of this study are as follows.

### 5.2. Theoretical Implications

First, this study contributed to explaining consumers' behavior toward eco-friendly hotels by identifying factors that affect customers' eco-friendly behavior. Although there have been various approaches to predict eco-friendly behavior in prior studies [34,107–111], a discussion on the management methods of hotel companies is needed as approaches to

verifying the relationship between customer attitudes and behaviors have been insufficient. This study therefore contributes to academic expansion by verifying the influence of eco-friendly hotel companies on CSR.

Second, in this study, the existing academic research area was expanded by considering the influence of the NAM on the behavior of customers with regard to eco-friendly hotels. In prior studies, there were attempts to explain complex decisions related to consumption behavior in eco-friendly hotels [20,43,55,108,109], although few studies have applied the NAM to predict customer behavior by using CSR of eco-friendly hotels as a cause variable. A number of previous studies verified the NAM as an efficient approach to predicting customers' eco-friendly behavior. Therefore, an academic contribution to future research has been made in relation to hotel companies and the relationship between the NAM, GCSR, and customer behavior.

Third, GCSR applied the TPB to predict eco-friendly behavior supported existing research results in predicting eco-friendly behavior of eco-friendly hotel users with the TPB. The study further verified the influence of the TPB SN on NAM PN, proving the correlation between the integrated approach of the TPB and the NAM. It was pointed out that the influence of the TPB should be increased in previous studies to predict eco-friendly behavioral intentions. This study proved, by showing a greater influence than previous research, that the integrated approach of the TPB and the NAM is useful for predicting customers' eco-friendly behavioral intentions.

### 5.3. Managerial Implications

First, eco-friendly hotels should continuously expose the management and activities of eco-friendly hotels to consumers from an ecological perspective. In other words, eco-friendly hotels should provide consumers with reasonable reasons for the inconvenience that eco-friendly activities cause or the price premium to be borne. A mid-to-long-term management strategy should be established so that hotel users can attain the psychological will to make sacrifices for the environment, and through this process, customers can form a positive attitude toward eco-friendly hotels [112]. Accordingly, public exposure to the management and activities of eco-friendly hotels together with customers' experiences can lead to positive behaviors such as premium price payment, revisitation to eco-friendly hotels, and making sacrifices for the environment.

Second, according to the research results that show consumers' AR, AC, and PN, in terms of the structural relationship in the NAM they are the leading factors of eco-friendly behavioral intentions for eco-friendly hotels, so these hotels should carry out marketing strategies to raise the level of AR, AC, and PN among consumers, which can be determined through the hotel's GCSR management strategy. Processes are needed to ensure that customers are aware of the hotel's GCSR activities and positive evaluation. These could include the following: (1) actively protecting nature by participating in community development through co-prosperity with eco-friendly start-up companies in the region that produce upcycled linen products; (2) running a campaign supporting children in crisis with dreams, blankets, and pillows through the funds (voluntary participation) of hotel senior management and executives; (3) performing GCSR by trading with start-up companies that consider the environment for the F&B part for the hotels to ensure that hotel companies are perceived positively.

### 5.4. Study Limitations and Future Research

First, the survey was conducted during the COVID-19 pandemic period. This did not consider the situation of respondents where government quarantine guidelines continued for more than two years. It may not have been easy for participants in the survey to clearly recall their experiences in eco-friendly hotels. It is also considered meaningful to compare and analyze people's perceptions by dividing the COVID-19 quarantine period in the future.

Second, the survey responses were gathered as a basic explanation for GCSR, but the questions that respondents might have had regarding the online survey could not be given immediate answers. In the future, when the COVID-19 pandemic has been alleviated, a direct study should be conducted with sufficient explanation of hotel companies and GCSR.

Third, responses were made by recalling the experience of eco-friendly hotels visited more than 1–2 times in the past two years. The CSR image of eco-friendly hotels visited by respondents may vary depending on the hotel brand, and the GCSR of luxury and upscale hotels may differ. Therefore, in future studies, it is advised that research is conducted with a wider and more appropriate sample. For example, limiting participation to only Marriott International users, or comparing and verifying Marriott vs. Hilton users, which may also be considered meaningful.

Fourth, this study is the result of examining the perceptions and attitudes of respondents according to the green hotel companies located in Korea. Specifically, in order to understand the perception of CSR images of the green hotel companies in Korea, various implications were presented based on the analysis results applying the TPB and NAM. However, green perceptions may vary depending on citizenship, culture, and economic level between countries. Therefore, it is expected that the limitation of respondents' representation in the hospitality industry will be expanded through other industries or various studies in the future.

**Funding:** This research received no external funding.

**Institutional Review Board Statement:** Not applicable.

**Informed Consent Statement:** Informed consent was obtained from all subjects involved in the study.

**Data Availability Statement:** The data are not publicly available due to privacy or ethical restrictions.

**Conflicts of Interest:** The author declares no conflict of interest.

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
