# Peer review of "A Study of the Integrated Model with Norm Activation Model and Theory of Planned Behavior: Applying the Green Hotel’s Corporate Social Responsibilities"

_sustainability, doi:10.3390/su15054680_

Round 1
Reviewer 1 Report
Introduction:
The title is not succinct and reflects the content of the article. It does provide insight into an important issue on how it gives impact to green hotels based on this case and customer behaviour. By shedding light, the insight is useful for managers who make decisions, particularly long-term strategy in any related green hotels. I believed other concerns are needed as the author highlighted to fill the research gap and contribute toward a better understanding of the green intention behaviour in association with other sources of values such as environmental issues not only CSR activities to provide a more comprehensive approach to consumer behavioural intention to patronage green hotels.
In the current business environment, the most valuable asset is company reputation and CSR has become one of the key components of corporate reputation. CSR is generally used to illustrate a business’s endeavours to achieve sustainable outcomes by committing to good business performance and standards. CSR is the ethical conduct of a company towards society, management acting responsibly in its relationships with other stakeholders and the ongoing commitment by a business to behave ethically and contribute to economic growth while developing the quality or class of life of the personnel as well as of the local community and society at large.
Assessing the green hotel usage behaviour in several studies indicate that a significant role of green trust in the adoption behaviour of consumers toward green hotels. The previous research suggests that the consumer green trust is related to their attitude and subjective norms, which are further associated with their behavioural consumption toward green hotels. While the perceived sustainable knowledge impacted the tourists’ subjective norms and attitudes which significantly affected their behavioural too.
For my point of view the author must also strongly argue for the significant problems/ issues - contributions in the hospitality industry. The introduction should be expanded to include some facts and figures about contributions to the business especially in the related green hotel business in Korea and the body of knowledge or theory building. Other related statistics to show why this is so important to study in terms of CSR and Green Behaviour as well as the linkages of social norm and personal norms. Why it needs to be studied?
Literature Review:
There were several bodies of literature on related research, the author is quite familiar with the existing state of research. The literature is still not very comprehensive Connections to prior work in the fields are made and served the article arguments that very clear in terms of distinctiveness of the study. Perhaps the author should discuss each of them concisely that allow a better understanding of the article. A clear explanation will provide a better view which is important to make this article interesting. The information and explanation given will help the green hotels/organization to sustain itself in the market.
The author failed to give a brief overview of the linkages of TPB and NAM. As known TPB provides a sound foundation for predicting consumers’ intentions and, subsequently, behaviour by analysing three belief categories: behavioural, normative and control beliefs. Behavioural beliefs are about the consequences of the target behaviour, as well as evaluations of those beliefs. Second, there are beliefs about the expectations of others (normative beliefs), such as family or friends, as well as a motivation to comply with the expectations of those people. Finally, there are beliefs about certain factors that might impede, or otherwise, the target behaviour (control beliefs), as well as the ability and desire to deal with them are considered many of the important points are missing in the LR as well as in the framework. Such as subjective norms are the person’s perceptions of what others think of a particular behaviour. Another element of the TPB, PBC acknowledges that one’s positive attitudes, or intentions, do not necessarily lead to action Specifically, PBC indicates people’s perceptions of their abilities to perform a given behaviour. I believed that author needs to explain and give justification to strengthen the framework and hypotheses.
Conceptual Framework
The author is trying to make it Interesting which opens a new arena in the green hotel’s consumption behaviour and CSR among tourists in Korea. The author has explained the underlying theory to support the framework however, the author failed to explain the linkage or relationship among each variable. For hypotheses, the author needs to provide a more depth argument and possible reasons the predicted relationship. I did not see an integrated model but only two models/theories that are combined.
Methodology:
The author should focus on the methodical thinking process – the key characteristics of values and its problem-solving. Generally, the methodology had not been discussed properly - on how it is executed, if possible, please add the details. The research design should clearly explain as should the reasons for selecting it, including its merits and limitations. In addition the questions based on CSR and other constructs should be formulated clearly and in such a way that all the study variables and their anticipated relationships had specified to provide clarity of the article. Just added more concrete explanation and simplified the discussion of the research approach and the justification for adopting the approach for this article. This will assist others to understand better.
Data Analysis:
The methods of analysis appropriate for the study design should be fully described even though the author is based on a case in Korea, but the author needs to justify, based on the objectives and the title given including their strengths and weaknesses. To ensure the practitioners can understand better, the author should also explain based on the objective of the article.
Findings & Discussion:
From my point of view, the literature was acceptable, but the overall discussion should be presented in a better way that allows other readers to understand – simple and clear. The connections to prior work in the field are also important, that assist other readers can compare with literature cited in the discussion. Please discuss further compared to previous related articles. Authors must act as marketers because I believed that this journal is also targeted to the practitioners. The authors can enhance the discussion on how the findings can assist hospitality organizations to enhance their product value and quality as well as develop trust among the current and potential customers in the current vulnerable environment. The customers who are more prone to environmental protection concerns have a more optimistic attitude, which further enhances behavioural consumption towards green hotels. Many researchers have suggested that the environmental concern of hotel customers influences their trust, which, in turn, contributes significantly toward their willingness to pay higher prices and their behavioural disposition toward green hotels. The discussion of the results and their implication as well recommendations are too superficial, and the author/s needs to add more because understanding values help in developing strategies. individual sense of belonging to a specific group or organisations as well as it is congruent with self-concept as well as values.
Reviewer 2 Report
The study investigates the topic of green hotels’ corporate social responsibilities integrating two theoretical models: Norm Activation Model and Theory of Planned Behavior. The paper presents interesting theoretical contributions and possible managerial contributions. However, I think is possible to further improve the following sections.
In the research methodology (section 3.1), the survey to verify the relationship between GCSR and AC, AR, PN, AT, SN, PBC, and GBI should be better clarified. Which are the content and the structure of the questionnaire?
In the discussion (section 5.1), results of the study should be analyzed and discussed more in-depth.
In limitations, does the study investigate all the dimensions of Sustainability? Environmental, Social and Economic?
Reviewer 3 Report
The paper is well written but some grammar points should be revised. I do recommend to make a new professional proofread review. At the same time, the conclusion and introduction should be updated according to the recently literature or the hot-released items recently published since 2021. Explain some sections easier or avoid to use jargon .. the following literature is suggested. Seraphin, H., & Korstanje, M. E. (2021). Neither Passive nor Powerless: Reframing Tourism Development in a Postcolonial, Post-conflict and Post-disaster Destination Context. Progress in Ethical Practices of Businesses: A Focus on Behavioral Interactions, 117-135.- Marpaung, B. O. Y., Aulia, D. N., & Witarsa, E. (2021). Evaluation of Tourism Policies Towards Sustainable Development. J. Pol. & L., 14, 1.-Thoradeniya, P., Lee, J., Tan, R., & Ferreira, A. (2015). Sustainability reporting and the theory of planned behaviour. Accounting, Auditing & Accountability Journal.-Patwary, A. K., Mohamed, M., Rabiul, M. K., Mehmood, W., Ashraf, M. U., & Adamu, A. A. (2022). Green purchasing behaviour of international tourists in Malaysia using green marketing tools: theory of planned behaviour perspective. Nankai Business Review International, 13(2), 246-265.
Reviewer 4 Report
Thanks for the opportunity to review your manuscript, and the overall quality of this study is OK. Following are my concerns.
Abstract: some abbreviations need to be explained when first come.
Introduction: a brief summary of the previous literature need to be discussed, and then identify the research gap.
Literature review
When you mentioned the previous study of both TPB and NAM in the green hotel context, a systematic review should be provided. A table is preferred to summarize the previous literature.
Method
First, you need to identify what the green hotel in Korea is, and how can you figure out if you actually targeted the participants who have ever stayed in the green hotels.
Why did you choose to use CFA rather than EFA?
As to the statistic results, the GFI=0.993 looks like pretty high, do you have any idea about it?
Table 1, some factor loadings are around 0.6, and one is below 0.6. Those loadings are a bit low.
Section 5, as you named is about the results and discussion, however, I did not find your discussion based on your results. However, you provided the implications in this section.
As to the implication, honest to say, the theoretical implication is relatively weak.
Minor concerns:
some sentences are dull and dry and repeat many times. Such as:
"........... interest in ........."
"it is necessary ........."
Format of the tables can be improved.
Round 2
Reviewer 4 Report
Thanks for your revising.
There is one more concern. Section 4. results, and section 5. Results and Discussion. There are some overlaps of those tiles.
Author Response
Thank you for your comment. I have corrected the part pointed out by the reviewer.